# Exposing the molecular heterogeneity of glycosylated biotherapeutics

Luis F. Schachner[1], Christopher Mullen[2], Wilson Phung[1], Joshua D. Hinkle[2], Michelle Irwin Beardsley [3], Tracy Bentley[3], Peter Day[3], Christina Tsai[3,8], Siddharth Sukumaran[3,9], Tomasz Baginski[3], Danielle DiCara[4], Nicholas J. Agard [4], Matthieu Masureel [5], Joshua Gober[6], Adel M. ElSohly[6], Rafael Melani[2], John E. P. Syka[2], Romain Huguet[2], Michael T. Marty[7] & Wendy Sandoval [1] ✉

The heterogeneity inherent in today's biotherapeutics, especially as a result of heavy glycosylation, can affect a molecule's safety and efficacy. Characterizing this heterogeneity is crucial for drug development and quality assessment, but existing methods are limited in their ability to analyze intact glycoproteins or other heterogeneous biotherapeutics. Here, we present an approach to the molecular assessment of biotherapeutics that uses proton-transfer charge-reduction with gas-phase fractionation to analyze intact heterogeneous and/or glycosylated proteins by mass spectrometry. The method provides a detailed landscape of the intact molecular weights present in biotherapeutic protein preparations in a single experiment. For glycoproteins in particular, the method may offer insights into glycan composition when coupled with a suitable bioinformatic strategy. We tested the approach on various biotherapeutic molecules, including Fc-fusion, VHH-fusion, and peptide-bound MHC class II complexes to demonstrate efficacy in measuring the proteoform-level diversity of biotherapeutics. Notably, we inferred the glycoform distribution for hundreds of molecular weights for the eight-times glycosylated fusion drug IL22-Fc, enabling correlations between glycoform sub-populations and the drug's pharmacological properties. Our method is broadly applicable and provides a powerful tool to assess the molecular heterogeneity of emerging biotherapeutics.

The increasing complexity of drug modalities is a growing challenge in the manufacturing and quality control of biotherapeutics[1]. Specifically, the composition of biomolecular modalities for potential drug candidates, including conjugated and fusion molecules, continues to expand beyond standard antibody formats, limiting the ability to control molecular heterogeneity during manufacturing[2,3]. Notably, many biotherapeutics entering the clinic, including IgG-based and Fc-fusion biologics, are glycosylated by design to ensure optimal

[1]Department of Microchemistry, Proteomics and Lipidomics, Genentech, Inc., South San Francisco, CA, USA. [2]Life Sciences Mass Spectrometry, Thermo Fisher Scientific, Inc., San Jose, CA, USA. [3]Pharmaceutical Technical Development, Genentech, Inc., South San Francisco, CA, USA. [4]Department of Antibody Engineering, Genentech, Inc., South San Francisco, CA, USA. [5]Department of Structural Biology, Genentech, Inc., South San Francisco, CA, USA. [6]Department of Protein Chemistry, Genentech, Inc., South San Francisco, CA, USA. [7]Department of Chemistry and Biochemistry, University of Arizona, Tucson, AZ, USA. [8]Present address: Protein Analytical Development, Ascendis Pharma, Palo Alto, CA, USA. [9]Present address: Translational Pharmacometrics, Janssen, Horsham, PA, USA. ✉e-mail: wendys@gene.com

therapeutic efficacy [4]. Glycans, carbohydrate moieties that modify more than 50% of the eukaryotic proteome, play a pivotal role in defining the pharmacological properties of biotherapeutics including potency, stability, bioavailability, solubility and immunogenicity [5–7]. Unlike nucleic acids and proteins, the biosynthesis of glycosylation is not template-driven. Instead, a complex network of metabolic and enzymatic reactions driven by genetic, epigenetic and environmental factors [8] extensively modify the glycan precursors, leading to structural diversification of the glycan chain termini [9]. For this reason, current analytical methods, particularly mass spectrometry (MS) [10–12], are challenged by the molecular heterogeneity that stems from protein glycosylation [10–12].

The heterogeneity of biotherapeutics, which often contain multiple glycosylation sites, is surpassing our ability to provide detailed molecular characterization without upstream enzymatic or chemical processing [3,13,14]. Current MS-based methods for glycoprotein analysis measure fragments of the biotherapeutic via released glycan composition or proteolytic glycopeptide analysis [15–17]. Likewise, attempts at measuring intact biotherapeutics have relied on the partial digestion of the glycans to reduce sample complexity [18,19]. Hence, targets for glycosylation-related critical quality attributes (CQAs) are typically limited to the extent of branching, galactosylation, sialylation, and terminal monosaccharide composition, providing an incomplete picture of the glycoform distribution or molecular heterogeneity of the biotherapeutic. These analytical obstacles pose important risks and challenges for biotechnology companies that need to implement 'quality-by-design' processes [1] and report human-compatible and consistent glycosylation, CQAs crucial for drug safety and efficacy [2,20].

To address this gap in molecular characterization, we turned to proton transfer charge reduction (PTCR) [21], an effective method for reducing mass-to-charge (m/z) spectral congestion that can occur during the MS analysis or fragmentation of complicated intact protein samples [22–27]. PTCR decreases the charge (z) of the ions inside the mass spectrometer, thereby expanding the m/z distribution of an initial ion population into product ions with charge states dilated over a larger m/z range (Supplementary Figure 1). When PTCR is applied to only a narrow m/z range (i.e., a subpopulation), the process provides a distribution of lower charge states at higher m/z for only that subset of ions, reducing spectral feature overlaps in the resultant spectrum (Fig. 1a-f). This in turn enables the observation of resolved peaks which correspond to the charge states of the protein analyte that are used to calculate its molecular weight [28]. Sequential application of PTCR to ion subpopulations via gas-phase fractionation [29] involves 'stepping' an isolation window over a suitably large m/z range, a form of a data-independent (DIA) tandem MS experiment, assuring that some charge states from all protein glycoforms of interest are subjected to charge reduction (Supplementary Figure 2). As a result, the DIA-PTCR workflow facilitates molecular weight measurement of all molecular forms of a biotherapeutic (i.e., proteoforms) [30,31] in a single intact MS experiment [28].

Application of DIA-PTCR to native glycoproteins is enabled by the development of the Orbitrap Ascend™ Tribrid™ mass spectrometer that permits the transmission and m/z isolation of increasingly larger biomolecules for ion-ion reactions. Although native MS [32] alone, which involves ionization of the protein analyte from nondenaturing solvent, has been applied to the characterization of glycosylated biomolecules and biotherapeutics, the resultant m/z spectra of large, heterogeneous targets are often uninterpretable due to the number of overlapping peaks in m/z space [19,33].

In this study, we utilize DIA-PTCR-generated spectra to carry out a qualitative analysis of the presence or absence of individual proteoforms within complex protein mixtures. This allows us to survey the full proteoform landscape of a biotherapeutic. By employing custom scripts executed in UniDec [34], we correlate DIA-PTCR data sets to information obtained from glycoproteomics and glycomics analyses.

This combination of data sources facilitates insight into factors beyond mere molecular weight distributions, such as the glycan composition of proteoforms.

Despite the intricacies of obtaining rigorous insights into the specific post-translational modifications of highly heterogeneous proteoforms, our work presents a bioinformatic strategy to approach this challenge. We use site-specific glycosylation data from bottom-up analyses and glycan composition data from glycomics analyses to decipher putative glycan compositions corresponding to observed proteoform molecular weights [30,31]. The outcomes of this analytical approach are graphically represented [35,36] as a monosaccharide fingerprint and a glycan barcode [37–39]. These visual tools delineate the monosaccharide composition of the biotherapeutic per molecular weight and the percent abundance of each glycan per site. Such characterizations of glycoproteins offer potential contributions to product characterization, in-process control, and GMP batch release testing for product-specific validation [40]. Ultimately, this strategy offers the potential to link the pharmacological effects of a biotherapeutic to specific glycoform sub-populations [41].

## Results

We first applied DIA-PTCR to profile the stoichiometry, chain pairing and glycoform distribution of a highly heterogeneous ligand fusion protein. This protein is a bispecific Fc-fusion construct assembled using 'knobs-into-holes' technology [42]. Each half contains three tandem copies of a murine Tumor Necrosis Factor superfamily ligand (TNF-L), each subunit containing one N-linked glycosylation site, fused to the N-terminus of the Fc domain (Fig. 1a). Additionally, to increase molecular specificity in vivo, a VHH domain [43] was fused to the C-terminus of one of the two Fc chains (Fig. 1a). This 'ligand hexamer' was exchanged into ammonium acetate solution (Fig. 1b) and delivered to the Orbitrap Ascend Tribrid mass spectrometer via native electrospray ionization (Fig. 1c). The full scan MS[1] spectrum (Fig. 1d) obtained under these conditions was spectrally dense due to the many overlapping ion signals and lacked discrete, distinguishable charge states.

To provide resolved charge states, 10 m/z-wide subpopulations of protein ions were quadrupole m/z selected via gas-phase fractionation (m/z selected as exemplified in Fig. 1e) and subjected to proton transfer charge reduction. This resulted in PTCR MS[2] spectra with m/z peaks corresponding to charge-reduced glycoform product ions dispersed over a wide m/z range higher in m/z than the isolation window (Fig. 1f; for detailed description of the DIA workflow and determining isolation width, see Supplementary Fig. 2a). The resultant set of acquired PTCR MS[2] spectra (Fig. 1g) were analyzed by UniDec using a 'sliding-window' approach which outputs both summed deconvolution results for combined spectra as well as deconvolution results for each individual spectrum of the experiment (Fig. 1h). With DIA-PTCR (Fig. 1i), we detected masses corresponding to the fully assembled and glycosylated molecule (at 175 kDa, Supplementary Figure 3) and to partial constructs missing the VHH domain (at 135 kDa). The masses at 115–119 kDa correspond to other partially assembled species, demonstrating that the analytical framework underpinning the DIA-PTCR method can resolve molecular heterogeneity stemming from glycosylation and other attributes, such as domain mis-assembly.

Recognizing the potential of the workflow as a broadly applicable method for the intact molecular characterization of complex biotherapeutics, we benchmarked reproducibility and accuracy using ovalbumin, a phosphorylated glycoprotein standard from chicken egg, resolving known glycoforms and co-occurring post-translational modifications [44] (Supplementary Figure 4). Over four replicate DIA-PTCR analyses, the dominant m/z ions and subsequent deconvolved neutral masses were found to be highly reproducible and consistent with those detected by intact MS analysis of the same sample (Supplementary Fig. 4d,e) [44].

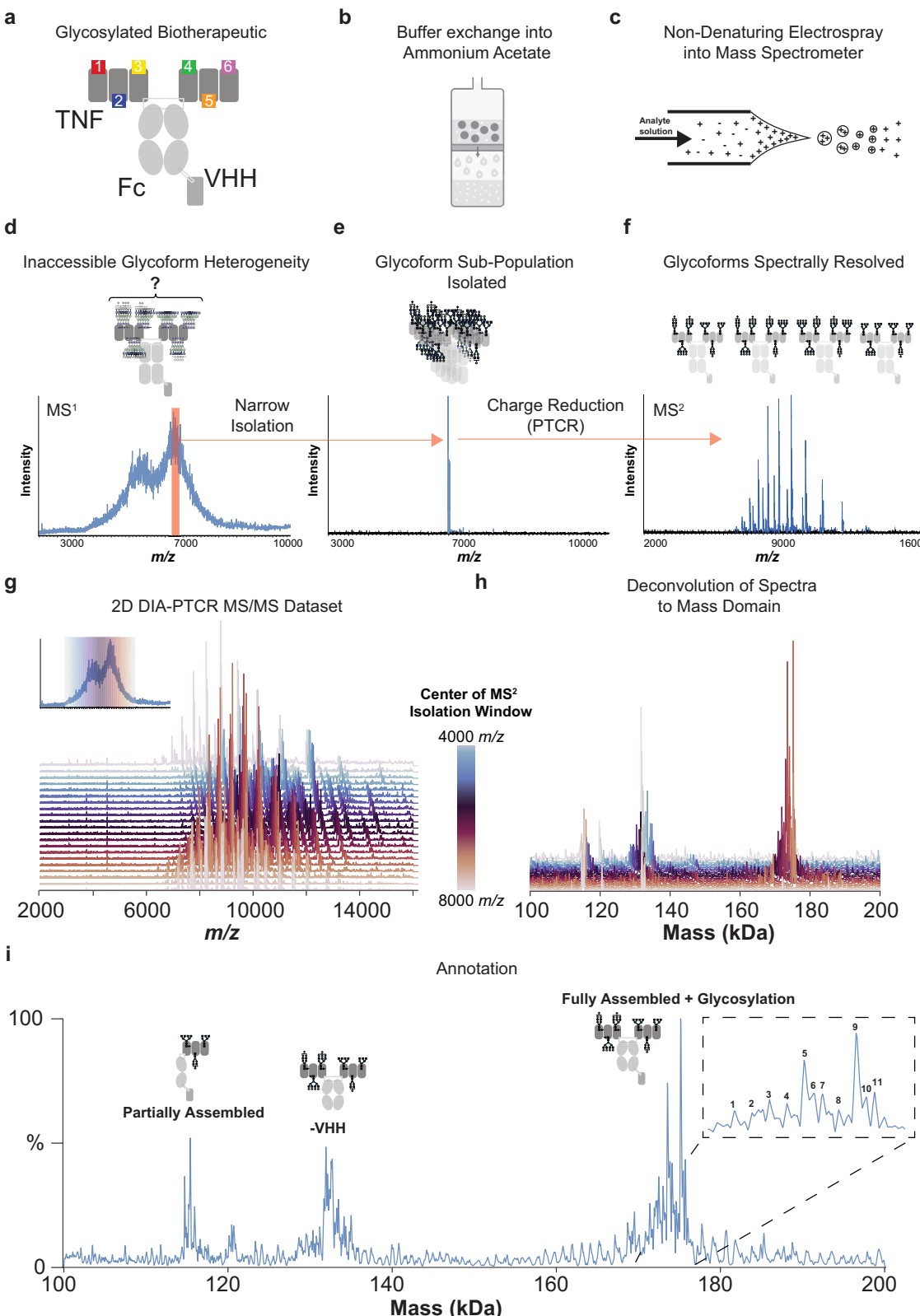

For a further proof-of-concept, the DIA-PTCR method was employed to profile the glycoform distribution of the peptide-bound Major Histocompatibility Complex class II DPA1*02:02/DPB1*05:01 allele (MHCII)[45,46], a non-covalent heteromeric complex which here-tofore could not be characterized intact by MS without fractionation due to the molecular heterogeneity arising from 4 N-linked glycosy-lation sites (inset of Fig. 2a). These molecules are important for cancer immunotherapy development, wherein patient-predicted neoantigen peptides (determined from patient tumor sequencing) are screened for binding against MHCII alleles using mass spectrometry. The experimentally validated peptide-MHCII complexes are then tested for a patient's T cell response to determine if the tested peptide is an effective cancer vaccine candidate[46]. Multiple N-linked and O-linked glycosylation sites on MHCII preclude direct MS[1] analysis of the

**Fig. 1 | DIA-PTCR workflow using ligand hexamer. a** Schematic of the ligand hexamer biotherapeutic with six glycosylation sites. **b** The glycoprotein is buffer-exchanged into ammonium acetate solution. **c** The protein is ionized by static nano electrospray (ESI) and introduced into the mass spectrometer. **d** Full scan MS[1] is first acquired to establish the *m/z* range of the analyte's ion signal. **e** Depiction of gas-phase fractionation, showing the mass spectrum of an isolated narrow *m/z* window containing only a subpopulation of the ions initially present in the MS[1] spectrum. **f** Proton transfer charge reduction on MS[2] isolation range (from **e**), producing a spectrum with resolved *m/z* peaks corresponding to consecutive

charge states of individual glycoforms. **g** Full stacked spectral representation of PTCR MS[2] spectra obtained by the DIA-PTCR method stepping the *precursor m/z* isolation window center through the entire *m/z* range of the biotherapeutic ions as detected in the MS[1] spectrum. **h** Neutral mass spectra obtained by mass deconvolution of each isolated PTCR MS/MS spectrum. **i** Composite mass deconvolution result after 'spectral stitching' of all *m/z*-to-mass deconvolved PTCR MS/MS spectra in the full DIA dataset using UniDec, with annotations elucidated in Supplementary Fig. 3.

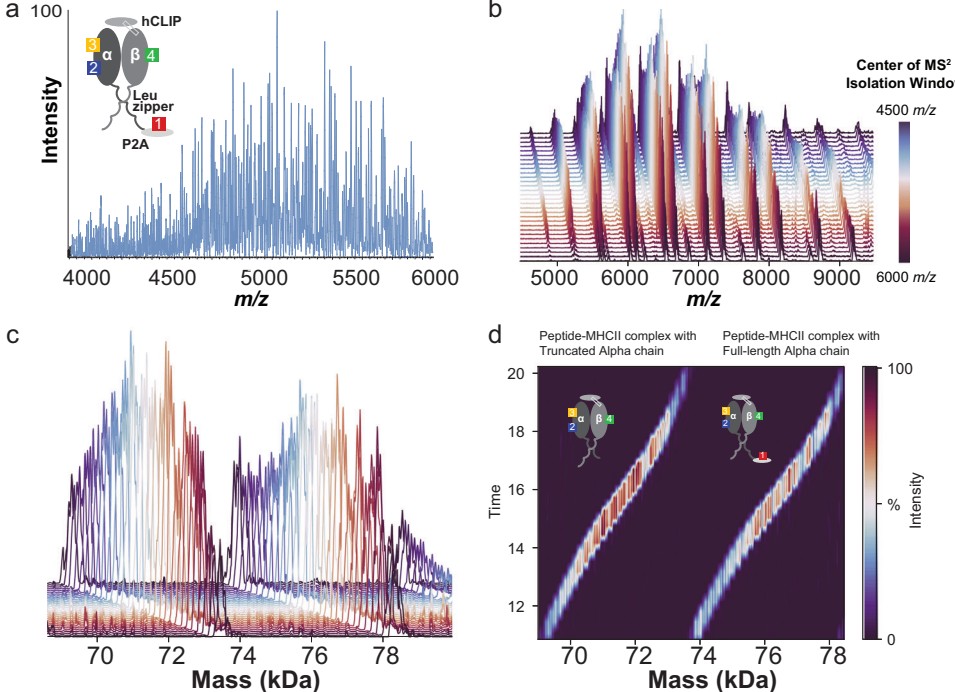

**Fig. 2 | DIA-PTCR on Major Histocompatibility Complex Class II shows truncated chain. a** Native *m/z* spectrum (MS[1]) of Major Histocompatibility Complex Class II DPA1*02:02/DPB1*05:01 allele, with inset showing schematic of MHCII heterodimer, with a human class II-associated invariant chain peptide (hCLIP), a Leucine zipper that promotes heterodimerization, and four glycosylation sites. Note that there is a non-native glycosylation site on the P2A peptide. **b** Entire

dataset PTCR-MS[2] spectra (stacked representation) from the DIA-PTCR analysis of the MHCII. **c** Entire dataset of PTCR-MS[2] spectra after mass deconvolution (stacked representation) from the DIA-PTCR analysis of the MHCII. **d** Heatmap representation of entire set of mass deconvolved PTCR-MS[2] (neutral) mass spectra, revealing two glycosylated populations corresponding to MHCII with truncated or full-length Alpha chain, with their respective schematic structures shown as insets.

---

peptide-bound complex due to spectral congestion (Fig. 2a). Hence, current methods denature and de-glycosylate the complex prior to analysis, focusing only on the released peptide[45,47]. With static nanospray ionization and the DIA-PTCR approach, we detected two populations of the intact, peptide-bound MHCII containing either a full-length or truncated alpha chain, each with distributions of masses consistent with extensive and heterogeneous glycosylation (Fig. 2b-d). The truncations within the alpha chain were validated by deglycosylation of the MHCII complex followed by partially denaturing reversed-phase LC-MS (Supplementary Figure 5).

As a final demonstration of the utility of DIA-PTCR, we sought to better understand the glycosylation profile of the IL22-Fc fusion biotherapeutic developed for epithelial repair (UTTR1147A), which consists of two IL22 cytokines dimerized by Fc domains that increase bioavailability[48–50]. The fusion protein contains 8 N-linked glycosylation sites, resulting in a highly complex MS[1] spectrum with no discernable *m/z* peak series with mass information (Fig. 3a). The observed level of molecular heterogeneity of this compound made data acquisition difficult on Orbitrap platforms with limitations in mass range (< 8000 *m/z*), isolation resolution (> 20 Th with ion trap isolation of *m/z* 2000–8000 precursors) and the space charge capacity attainable with linear ion trap isolation at low Mathieu *q* values. These limitations

necessitated a migration of our efforts to the Orbitrap Ascend platform to explore the extended *m/z* range (up to *m/z* 16,000) to more fully utilize PTCR charge reduction. (Supplementary Fig. 6).

We modified the Orbitrap Ascend (for research purposes) to extend the quadrupole *m/z* filter precursor selection range to 8000 *m/z* by lowering the drive frequency, and compared the performance in this extended mass range mode to precursor isolation in the RF linear ion trap analyzer (Supplementary Fig. 7). Employing the high *m/z* quadrupole precursor selection proved advantageous in decreasing the number of proteoforms present in the individual stepped *m/z* selection windows relative to ion trap *m/z* isolation and provided more interpretable MS[2] spectra with higher product ion signal to noise ratio and less deconvolution ambiguity. In the analysis of other heterogeneous biotherapeutics over the past year we assert that inclusion of the higher resolution quadrupole precursor *m/z* selection into the workflow is absolutely necessary for mass interpretation of spectrally dense data.

To analyze the IL22-Fc constructs we utilized the 16,000 *m/z* mass range and a 20 Th quadrupole isolation window. DIA-PTCR detected >160 masses for IL22-Fc from a single analysis (Fig. 3b, c). The IL22-Fc molecule was compared before and after sialidase treatment, resulting in removal of the outermost sialic acid moieties that contribute to the

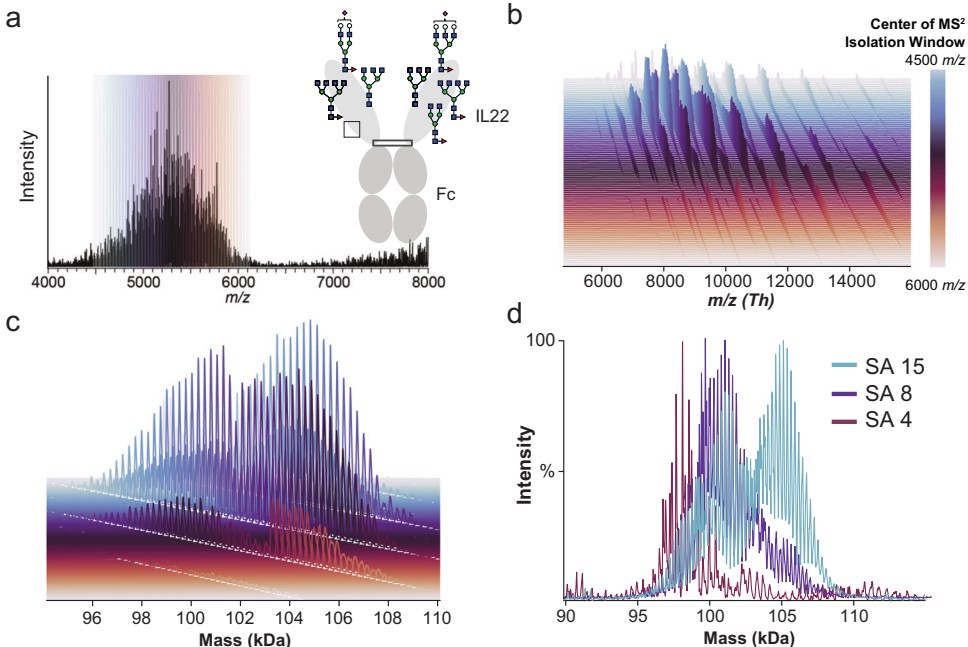

**Fig. 3 | Glycoform heterogeneity of an Fc fusion protein is resolved by DIA-PTCR. a** Native full-scan MS of IL22-Fc, with inset showing structure and eight potential glycosylation sites of the biotherapeutic. The specific glycoform shown, with only seven sites occupied, is elucidated in Supplementary Fig. 15. **b** Overlaid mass spectra following DIA-PTCR. **c** Overlaid deconvolution results of the mass spectra in (**b**). **d** Glycoform mass distributions for IL22-Fc samples enriched for 4, 8 and 15 mol/mol sialic acid content.

proteoform mass diversity. An overall mass reduction of 6000 Da was observed, indicating both near complete removal of the twenty-two sialic acids as well as highlighting the ability of the DIA-PTCR workflow to detect changes in proteoform heterogeneity from one sample to the next (Supplementary Fig. 8). As an additional validation of the reproducibility of the DIA-PTCR workflow from data acquisition to mass deconvolution of complex biotherapeutics, we analyzed the sialidase-treated IL22-Fc in three technical replicates. This experiment produced three equivalent distributions of molecular weights with Pearson correlation coefficients of 0.95–0.98 (Supplementary Fig. 9).

Finally, we were interested in testing the ability of the DIA-PTCR method to assess batch-to-batch reproducibility. As a surrogate for multiple batches of the Fc fusion protein, given that only a single large-scale purification was available, we fractionated and enriched IL22-Fc samples for 4, 8 and 15 mol/mol sialic acid molar content (Fig. 3d, Supplementary Fig. 10). We detected extensive differences in the molecular weight distributions of the three samples that were consistent with increasing amounts of sialylation.

### Application of DIA-PTCR to the assignment of intact glycoforms

Having obtained a distribution of molecular weights that correspond to putative glycoforms of both peptide-bound MHCII and IL22-Fc, we were interested in the potential to annotate them with corresponding glycan structures. Nonetheless, simply matching each of the N-linked sites to known glycans from a database of species-specific candidates yielded multiple alternate assignments for each observed molecular weight – a well-known inference obstacle in the interpretation of glycoforms (Supplementary Fig. 11a).

Our solution involved associating these molecular weights with compositions of simple saccharides or monosaccharides (such as sialic acid, hexose, GlcNAc, and fucose). This method provided a simplified characterization of the glycoprotein's modifications. However, a multitude of possible combinations of these monosaccharides made these distributions broad, and consequently, lowered the precision of asserting the number of monosaccharides for each glycoform (Fig. 4b, Supplementary Fig. 11).

We next employed glycoproteomic data to complement the DIA-PTCR data in reducing ambiguity of monosaccharide assignments. This pairing narrowed down our database of possible glycans, allowing us to assign glycan structures with greater confidence. This methodology change led us to find consistent monosaccharide compositions in most MHCII glycoforms. We found that 100% of MHCII glycoforms contained 1 fucose, 90% contained 7 or 8 units of GlcNAc, and 75% contained 1 or 2 sialic acids (Supplementary Fig. 11). Notably, the hexose content distribution was wider than that for other monosaccharides, ranging from 2-8 units. Taking this workflow one step further, we used whole glycan structures from our glycopeptide analysis to create a barcode as a visualization tool that provided detailed information about the makeup of the MHCII sample, including predictions of relative abundances of glycoforms (Fig. 4c, Supplementary Fig. 11d).

Assigning glycan structures to IL22-Fc, however, presented a tougher challenge due to its eight glycosylation sites, leading to a vast assortment of possible combinations with over 180 candidate glycans. As was the case with MHCII, combining the molecular weight information from DIA-PTCR with glycoproteomic data gave us the most accurate potential assignments (Fig. 4d–f, Supplementary Fig. 12). This strategy confirmed the expected increases in sialic acid content, displayed constant levels of GlcNAc, a slight increase of fucose content across the three conditions, and revealed an increase in hexose content concomitant with sialic acid. An orthogonal glycomic analysis of these samples (2AA-HILIC, Supplementary Fig. 13), reiterated the presence of consistent fucosylation across samples, corroborated an increase in sialic acid, and attributed the observed increase in hexose content to an increase in galactose and in tetra-antennary structures.

Importantly, the monosaccharide fingerprint for IL22-Fc revealed these trends in the sialic acid and hexose content that were still apparent even without constraining the monosaccharide search space, illustrating that critical results and changes between samples may be reproducibly observed even without companion glycoproteomic information or other data from orthogonal analyses (Fig. 4d–f, Supplementary Fig. 14).

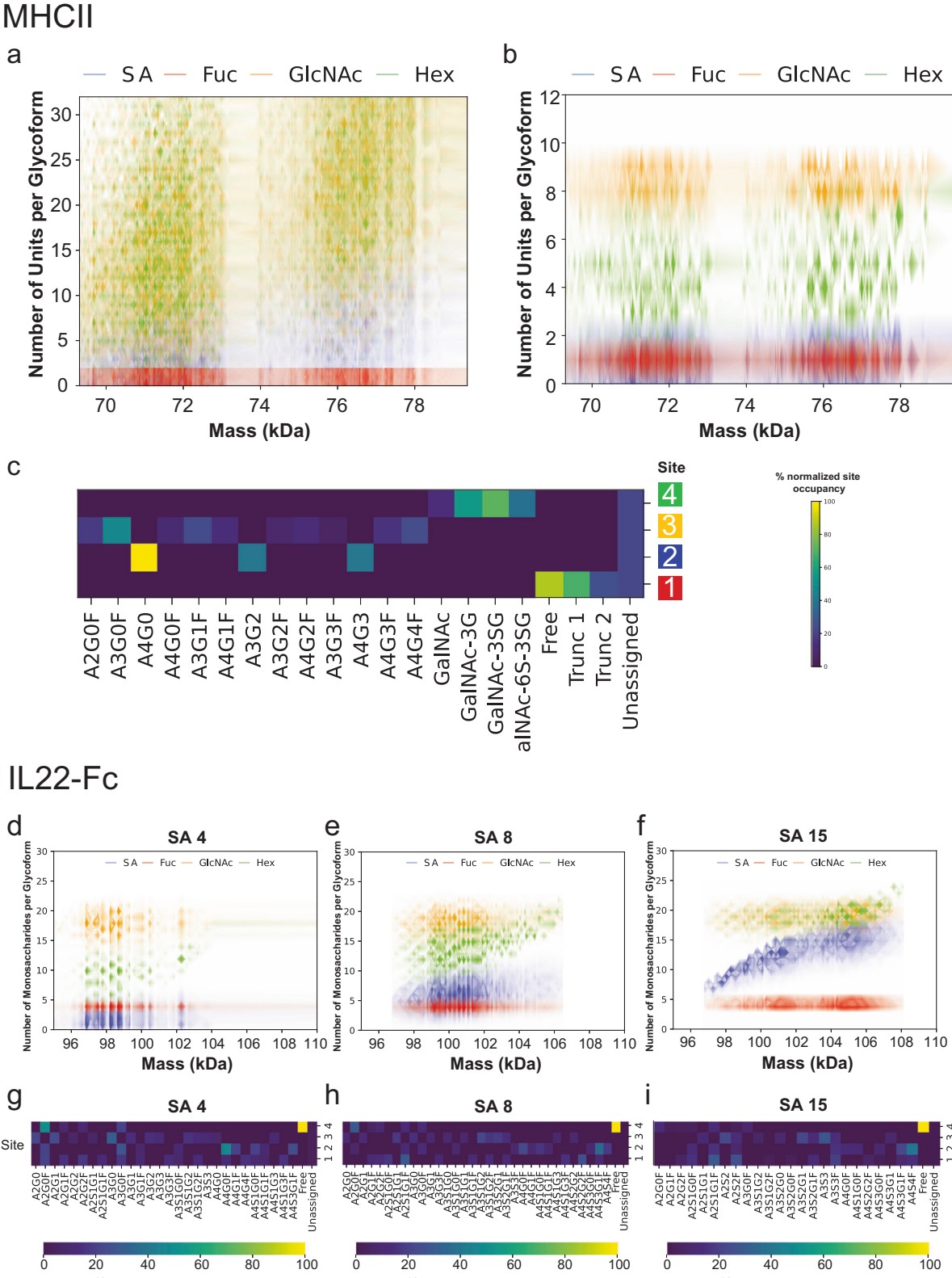

Despite the complexity of the IL22-Fc samples, the glycan barcodes (Fig. 4g–i, Supplementary Figs. 15, 16, Supplementary Data 2–4) successfully mapped IL22-Fc glycosylation and predicted the relative abundances of glycoforms. However, the inherent glycan micro-heterogeneity, such as the potential for dozens of modifications at each glycan site, resulted in thousands of possible glycoforms with the same mass, making it challenging to definitively assign glycoforms to individual weights. To exemplify this point, the most abundant peak in the IL22-Fc SA4 sample with a mass of 98,224 Da could be attributed to 614,270 potential glycoform species within a window of ±5 Da. The most abundant species out of all possible glycoforms for this molecular weight was calculated to have an abundance of 0.034% and is depicted in the inset of Fig. 3a (Supplementary Fig. 15).

**Fig. 4 | Profiling glycosylation on glyvosylated biotherapeutics using orthogonal datasets. a** Glycoform-resolved monosaccharide fingerprint of MHCII using DIA-PTCR data only, showing number of monosaccharides per intact glycoform mass, with glycoform probability encoded by the transparency of the color. **b** Glycoform-resolved monosaccharide fingerprint of MHCII using integrated DIA-PTCR and glycopeptide data, showing number of monosaccharides per intact glycoform mass, with glycoform relative abundance encoded by the transparency of the color. **c** Glycan barcode for MHCII based on *N-* and *O-* glycan structures determined by glycopeptide analysis, using quantitative input from the integrated approach. The barcode denotes the normalized relative abundance of a glycan assignment per site, color coded according to schematic in (**a**). On the barcode, the first site corresponds to the non-native site on P2A and the fourth site denotes an O-linked glycosylation site. **d**–**f** Glycoform-resolved monosaccharide fingerprints for IL22-Fc at SA 4, 8 and 15, upon integration with glycoproteomic data, showing number of monosaccharides per intact glycoform mass, with glycoform abundance encoded by the transparency of the color (Supplementary Figs. 12, 14). **g**–**i** Glycan barcodes for IL22-Fc SA 4, 8 and 15 reflecting the normalized relative abundance of a glycan assignment per site (Supplementary Fig. 16).

## Correlating glycoform subpopulations to a biotherapeutic's pharmacological properties

We next sought to correlate putative IL22-Fc glycoform subpopulations (Fig. 5a, b) with the biotherapeutic's sialic-acid dependent pharmacological properties. Interestingly, we found that the binding of IL22-Fc to its receptor was inversely correlated with increasing sialic acid content (Fig. 5c) and that enzymatic cleavage of IL22-Fc sialylation led to a near two-fold increase in binding to the IL22 receptor as measured by ELISA (Fig. 5d). This decrease in potency of the biotherapeutic may be largely ascribed to the highly sialylated IL22-Fc glycoforms. We hypothesized that large, highly sialylated glycans can potentially interfere with the cytokine-receptor interface and decrease receptor binding [51]. In fact, examination of the published structure of IL22 bound to its receptor showed that Asn21 sits at the cytokine-receptor interface (Fig. 5f–k, Supplementary Fig. 17). Hence, to assess the potency contribution of individual IL22-Fc glycosylation sites, we knocked out each site in turn by point-mutation, finding that the N21Q mutation led to a 3.98-fold increase in potency relative to wild type (WT) (Fig. 5e).

## Discussion

We established a platform for fast direct compositional analysis of highly complex biotherapeutics that heretofore have eluded intact MS characterization. DIA-PTCR enabled investigation into the molecular heterogeneity of biotherapeutics by displaying a detailed landscape of the proteoforms present in the sample. Using this approach, we observed subunit truncations, differential glycosylation between samples, and were able to obtain mass ranges which enabled confirmation of subunit assembly in the midst of extreme proteoform diversity, highlighting the 'something from nothing' nature of this simple analysis. As exemplified by the eight-times glycosylated Fc fusion protein, we identified distributions of masses consistent with the glycoforms of IL22-Fc and of other challenging biotherapeutics in a single experiment without denaturation, upfront digestion or separation, retaining information about non-covalent structure, co-occurrence of glycans and other PTMs such as phosphorylation and acetylation.

The gold standard for industrial glycoprotein characterization is the 2AA-HILIC glycomic analysis, a highly time-consuming workflow that is performed multiple times during a biotherapeutic's development and manufacturing. In contrast to these standard multi-step glycomic analyses, the DIA-PTCR workflow requires minimal sample manipulation, provides results in minutes and may be fully automated. Moreover, the inherent richness of the DIA-PTCR datatype is exemplified by its ability to define both a biotherapeutic's structure – such as mis-assembled constructs in the 'ligand hexamer' or a truncated alpha chain in peptide-bound MHCII – and the overall glycoform distribution in a single experiment. While the 2AA-HILIC method and related glycomic and glycopeptide approaches provide in-depth information about the sample's glycan composition, these current workflows are blind to the actual glycan combinations that exist in the biotherapeutic sample as encoded in a proteoform's intact molecular weight. The DIA-PTCR method specifically addresses this gap in knowledge by providing an unprecedented window into the molecular weights and overall distribution of the proteoforms in heterogeneous biotherapeutic samples. Moreover, we demonstrate that general trends and broad-scale deviations in the monosaccharide compositions of IL22-Fc SA variants can be ascertained using DIA-PTCR. However, more precise glycan assignments for comprehensive glycoform profiling require complementary methods to further constrain the bioinformatic search space.

Due to the staggering combinatorial diversity of glycosylation, probable annotations for the detected intact masses – which reflect the limited set of glycan combinations present in the sample – were modeled using validated glycan candidates from orthogonal approaches. As stated, the monosaccharide fingerprint is generated after bioinformatic integration of the DIA-PTCR data with glycopeptide and glycomic data. Based on the experimentally determined abundance of each glycopeptide, and upon validation of specific glycans using glycomic chromatographic information, a statistical model was formulated to calculate the relative abundance of defined monosaccharide and glycan compositions for each DIA-PTCR molecular weight. While the model assumes that each glycosylation site is independent from modification at other sites, knowledge of glycobiology can be incorporated in future applications to reflect any potential crosstalk between modification sites. Overall, this approach increased the sensitivity of the DIA-PTCR analysis to changes in the structure and glycoform landscape of the glycosylated biotherapeutic that may prove useful in quality assessment during and after manufacturing.

We found that the utility of this compositional information was a function of the actual molecular complexity of the glycoprotein. That is, the more heterogeneous the glycosylation pattern on a protein, the more complex the mixture of glycoforms for a given molecular weight. For instance, the annotation of MHCII glycoforms was tractable as there were only 360 predicted glycoforms. In contrast, the extensive heterogeneity of IL22-Fc – which was modeled to have $>2.38 \times 10^8$ possible glycoforms after constraining the search space – resulted in very low abundances for even the most abundant glycoform assignments per molecular weight (i.e., >600,000 possible glycan combinations for the proteoforms weighing 98,224 Da). Nonetheless, even for the most complex of samples, the DIA-PTCR platform can still expose, in the form of intact mass distributions, a biotherapeutic's molecular heterogeneity.

As a demonstration of the possible applications of the integrated DIA-PTCR datatype beyond industrial quality assessment, we correlated pharmacological effects, namely the potency of the biotherapeutic, to specific IL22-Fc glycoform subpopulations, defining a general path toward profiling glycoform distributions and the structure-function annotation of glycoforms. This level of molecular precision for the description of glycoproteins may aid in the direct interrogation of glycoform-specific effects, such as those mediated by glycan-receptor interactions, wherein small differences in the glycan structures may affect recognition by the reader[52]. With glycoform resolution, and coupled with glycoengineering approaches or synthetic glycoform reagents[53–56], glycan signaling may be studied in the context of the whole polypeptide, which considers contributions from all glycosylation sites and other co-occurring PTMs[57,58]. Moreover,

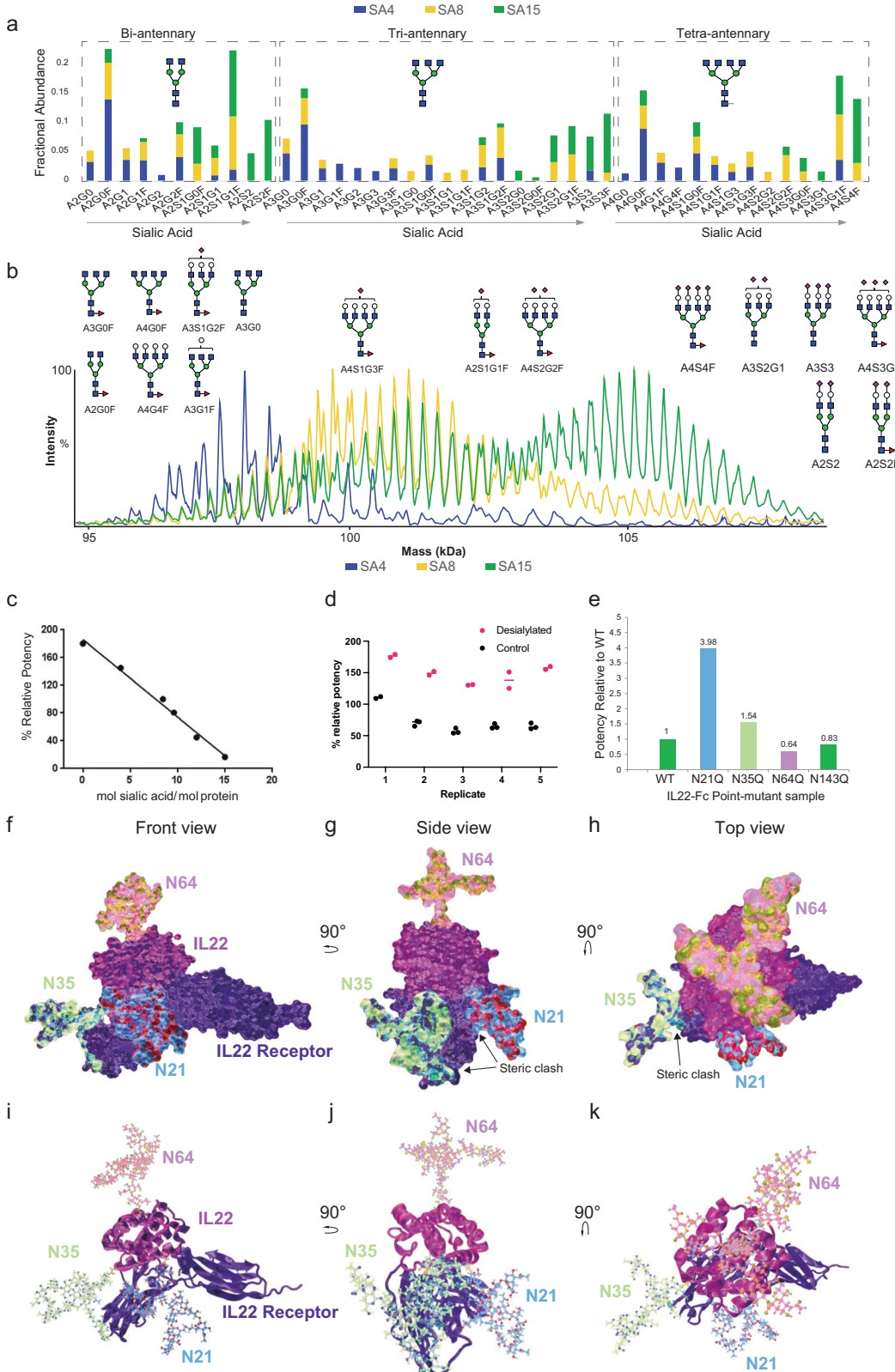

Fig. 5 | The sialic acid states and structures of IL22-Fc. a Comparison of glycan structures and their relative abundance detected in IL22-Fc SA4 (blue), 8 (yellow) and 15 (red) samples. b Glycoform mass distributions for IL22-Fc samples enriched for 4, 8 and 15 mol/mol sialic acid content, with annotations denoting the name and structure of representative glycans comprising abundant IL22-Fc glycoforms. c Relative potency of IL22-Fc measured as a function of sialic acid content. d Percent of relative binding of control or enzymatically desialylated IL22-Fc to its receptor, IL22R1A, as measured in duplicate by ELISA. e Relative potency of IL22-Fc point mutants and wild-type (WT) at each glycosylation site measured in duplicate. f–k Structure of glycosylated IL22 bound to its receptor (PDB: 3DLQ) visualized from multiple perspectives in ribbon and surface representations, showing the location of the IL22 glycosylation sites (N21, N35 and N64) and some steric clashes between large representative glycans (A4S4) and the IL22 receptor as a result of their proximity to the cytokine-receptor interface. A structural model in augmented reality is available in Supplementary Fig. 17.

glycoform annotation may contribute to more accurate modeling of the glycan density on a protein, which may affect its aggregate biophysical properties[59,60].

Future development of the DIA-PTCR workflow may incorporate top-down fragmentation of intact masses for glycoform characterization, affording further constraints to the glycan annotation in a single intact MS experiment. We also foresee leveraging this approach to analyze immuno-enriched glycoproteins to elucidate the network of readers, writers and erasers of glycosylation.

## Methods

### Sample preparation

**Generation of 'Ligand hexamer'.** Samples were generated by transient transfection of in house CHOK1 cells. Conditioned media was collected after ten days, and the recombinant proteins were purified using a combination of affinity chromatography and size exclusion chromatography.

### Generation of MHCII sample

**Construct design.** The protein sequence of DPA1*02:02/DPB1*05:01 was obtained from hla.alleles.org. The genes for the extracellular domains of the MHCII alpha chain and beta chain were codon optimized and cloned into a mammalian expression vector under the control of a CMV promoter. A P2A self-cleaving peptide sequence was inserted between the alpha chain and beta chain to create a bicistronic expression vector. A signal sequence for secretion was included before the N-terminus of the alpha chain, and the Fos zipper was added to the C-terminus of the MHCII alpha chain extracellular domain followed by an Avi tag (GLNDIFEAQKIEWHE) and a hexahistidine tag. The MHCII invariant chain peptide (CLIP) was fused to the N-terminus of the beta chain using a flexible glycine/serine linker. A signal sequence for secretion was included before the CLIP peptide, and the Jun zipper was added to the C-terminus of the beta chain extracellular domain followed by a hexahistidine tag.

HHV1P-gD-SS_DPA1*02:02-linker-thrombin-Fos-linker-Avi-His6_-P2A_HHV1P-gD-SS_linker-CLIP-linker-TEV-linker-DPB1*05:01-linker-thrombin-Jun-linker-SpyTag003-linker-His6. [viralsigseq].

GAIKADHVSTYAMFVQTHRPTGEFMFEFDEDEQFYVDLDKKETVWH LEEFGRAFSFEAQGGLANIAILNNNLNTLIQRSNHTQAANDPPEVTVFPKEP VELGQPNTLICHIDRFFPPVLNVTWLCNGEPVTEGVAESLFLPRTDYSFHKF HYLTFVPSAEDVYDCRVEHWGLDQPLLKHWEAQEPIQMPETTESSADLVP RGSLTDTLQAETDQLEDEKSALQTEIANLLKEKEKLEFILAAGGSGGSGLN DIFEAQKIEWHEHHHHHHGSGATNFSLLKQAGDVEENPGP[viralsigseq] GDSGTPVSKMRMATPLLMQAGGGGSENLYFQGGGGGSRATPENYLFQ GRQECYAFNGTQRFLERYIYNREELVRFDSDVGEFRAVTELGRPEAEYW NSQKDILEEKRAVPDRMCRHNYELDEAVTLQRRVQPKVNVSPSKKGPLQ HHNLLVCHVTDFYPGSIQVRWFLNGQEETAGVVSTNLIRNGDWTFQI LVMLEMTPQQGDVYICQVEHTSLDSPVTVEWKAQSDSARSKSSADLVPR GSRIARLEEKVKTLKAQNSELASTANMLREQVAQLKQKVMNHGGSGGS RGVPHIVMVDAYKRYKGGSHHHHHH.

**Protein expression and purification.** The His-tagged MHCII proteins containing C-terminal Fos/Jun leucine zippers were expressed in Chinese Hamster Ovary (CHO) cells. The proteins were purified by Nickel-NTA chromatography followed by size exclusion chromatography. Protein purity was determined to be >90% by SDS-PAGE.

**Generation of IL22-Fc SA variants and point-mutants.** DIA-PTCR analyses were performed on IL22-Fc sialic acid variants 4, 8, and 15 N-acetylneuraminic acid (NANA) mol/mol. The sialic acid variants were prepared from 10 and 16-day cell culture fermentations (to control for sialylation) followed by centrifugation, affinity column (MabSelect Sure, p/n17-5438-01, Cytiva USA) and sulphopropyl (SP) Sepharose Fast Flow (Sigma-Aldrich, St Louis, MO) ion exchange chromatography with fractionation to enrich for sialic acid content. All sialic acid

variants were formulated in 10 mM sodium phosphate, 240 mM sucrose, 0.02% Polysorbate 20, 5 mM methionine, pH 7.1, at a final nominal concentration of 10 mg/mL Il22-Fc.

The sialic acid reversed-phase high-performance liquid chromatography (RP-HPLC) method was used to determine N-acetylneuraminic acid (NANA) content in IL22-Fc. Sialic acids were released via acid hydrolysis, derivatized with o- phenylenediamine (OPD), and analyzed by C-18 RP-HPLC with fluorescence detection. The sialic acid concentration was calculated from an external calibration curve and reported as moles of NANA per mole of IL22-Fc.

**Separation of IL22-Fc glycoforms by isoelectric focusing.** A purity method based on imaged capillary isoelectric focusing (ICIEF) was developed for the separation of IL22-Fc glycoforms. The separation was found to be driven in part by the number of negatively charged sialic acids attached to each glycoform. The ICIEF glycoform charge assay was used to provide quantitative information on the distribution of sialic acid in IL22-Fc. IL22-Fc samples were treated with CpB to remove C-terminal lysine charge heterogeneity, mixed with an ampholyte solution containing a final concentration of 8 M urea to denature the protein, and separated in a capillary cartridge using a ProteinSimple iCE3 imaged cIEF system.

**N-linked Glycan Analysis by 2-AA HILIC UHPLC.** The hydrophilic interaction chromatography (HILIC) method was used to determine the relative distribution of N-linked glycans in IL22-Fc. N-linked glycans were enzymatically released using PNGase F, derivatized with 2-aminobenzoic acid (2-AA), and analyzed by HILIC ultra-high performance liquid chromatography (UHPLC) with fluorescence detection. HILIC is a variation of normal phase chromatography in which retention is due to the size and polarity of glycans. The early eluting peaks represent smaller, neutral glycans and the late eluting peaks represent larger, sialylated glycans. As sialic acid content increases, an increase in number of peaks and peak areas for the larger, sialylated glycans is observed along with a corresponding decrease in number of peaks and peak areas for the smaller, neutral glycans.

**Generation of IL22-Fc Asn point-mutants.** Asn to Gln point-mutations were introduced to IL22-Fc residues N21, N35, N64 and N143 and were prepared from 10 day CHO cell culture fermentations followed by sulphopropyl (SP) Sepharose Fast Flow ion exchange chromatography. All mutants were formulated in 10 mM sodium phosphate, 240 mM sucrose, 0.02% Polysorbate 20, 5 mM methionine, pH 7.1.

**Generation of Ovalbumin.** Ovalbumin sample was purchased from Thermo Fisher Scientific (77120).

**Sample preparation for intact MS DIA-PTCR analysis.** Samples at a concentration of 10 μM were exchanged into 200 mM ammonium acetate solution using BioRad Bio-Spin 6 filters (7326227) or Zeba™ Spin Desalting Columns, 7 K MWCO, 0.5 mL (89882, Thermo Fisher Scientific).

**N-linked Glycosylation Site Mapping (LC-MS/MS analysis).** Proteins were reduced with 20 mM dithiothreitol for 30 min at 37 °C followed by alkylation with 40 mM iodoacetamide at room temperature for 30 min. To obtain sequence coverage for all potential N-linked glycosylation sites, each sample was digested overnight at 37 °C with one or more of the following enzymes separately: trypsin (Promega), Lys-C (Roche), Glu-C (Roche), and chymotrypsin (Thermo Fisher Scientific). Digests were quenched with 0.1% TFA and subjected to C18 stage-tip clean up with a 40% acetonitrile, 59.9% water, 0.1% TFA elution step. After cleanup, peptides were dried down and reconstituted in 50 μL 0.1% TFA, where 1 μL was injected using an UltiMate™ 3000 RSLCnano

system (Thermo Fisher Scientific) via an autosampler and separated on a 35 °C heated Aurora UHPLC Emitter C18 column (75 μm × 25 cm, 120 Å, 1.6 μm resin, IonOpticks). A binary gradient pump was used to deliver solvent A (97.9% water, 2% acetonitrile and 0.1% formic acid) and solvent B (97.9% acetonitrile, 2% water and 0.1% formic acid) as a gradient of 2% to 35% solvent B over 40 min at 0.3 μL/min. The solvent was step-changed to 75% solvent B over 5 min and then held at 90% for 5 min to clean the column. Finally, the solvent was step-changed to 2% solvent B and held for 10 min for re-equilibration. Separated peptides were analyzed on-line via nanospray ionization into an Orbitrap Fusion™ Lumos™ Tribid™ mass spectrometer (Thermo Fisher Scientific) using the following parameters for MS1 data acquisition: 240,000 Orbitrap resolution; 350–1350 m/z scan range; 30% RF lens; 250% normalized AGC target; 50 ms maximum injection time; 1 microscan; centroid data type; positive polarity. The following parameters were used for MS2 data acquisition: 0.7 m/z quadrupole isolation window; HCD activation with 30% CE; 200–1200 m/z scan range in the ion trap; 200% normalized AGC target; 11 ms maximum injection time; 1 microscan; centroid data type. Data was collected in cycle time data dependent mode with a dynamic exclusion of 1 for 10 s. Detected peptides were searched against the protein sequences using Bio-Pharma Finder 5.0 Peptide mapping (Thermo Fisher) and PMI Byologic (Protein Metrics Inc.).Glycopeptide search results are provided as Supplementary Data.

**Intact MS DIA-PTCR - Mass Spectrometry methods.** Mass spectra were acquired on a Thermo Scientific Orbitrap Eclipse™ or Orbitrap Ascend™ mass spectrometer equipped with the high m/z range HMRn (8k m/z range Eclipse, 16k m/z range Ascend) and Proton Transfer Charge Reduction (PTCR) options. In addition, the Orbitrap Ascend™ was modified (for research purposes only) to enable quadrupole isolation to 8k m/z to better understand the influences of quadrupole isolation on the outcome of the DIA-PTCR m/z spectra, a modified Orbitrap Ascend™ with a low frequency quadrupole was used to enable high mass quadrupole isolation. The PTCR reagent, perfluoroperhydrophenanthrene, was delivered at a regulated flow for anion production. Tandem mass spectrometry experiments were accomplished using static nanospray and utilized ion trap isolation with an isolation widths of (10–20 m/z (Ascend™ quadrupole isolation), and 50–100 m/z (Eclipse™ and Ascend™ ion trap isolation). PTCR reaction times varied from 5–100 ms depending upon the amount of charge reduction (mass dispersion) required. Both the isolation width and the reaction time were manually optimized to give spectra that dispersed the product ion population throughout a good majority of the available instrument m/z range, and yielded spectra with clearly resolvable peaks. In general, as the sample heterogeneity increases, the isolation window should be decreased so that the number of proteoforms in any isolation window is reduced, and product ion dispersion is maximized. The instruments were operated in intact protein mode (high pressure) and mass spectra were acquired in the Orbitrap using averaging of 100 and a resolution of 7000 at m/z 200.

**IL22-Fc potency by binding assay.** The IL22-Fc potency assay measured the ability of IL22-Fc to bind to the IL-22 RA1 extracellular domain (ECD). In the assay, varying concentrations of IL22-Fc reference standard, control, and samples are added in duplicate to a 96-well plate coated with IL-22 RA1 ECD. Bound IL22-Fc is detected with goat anti-human IgG-horseradish peroxidase (HRP) antibody (c/n AP112P, Sigma-Aldrich, St Louis, MO) and a 3,3′,5,5′-tetramethylbenzidine substrate solution (p/n T4444, Sigma-Aldrich, St Louis, MO). The mean of the duplicate results, expressed in optical density (OD) units, are plotted against IL22-Fc concentrations, and a parallel curve program is used to calculate the measured potency of IL22-Fc samples relative to the reference standard.

**Molecular modeling of glycosylated IL22 bound to IL22 receptor.** The PDB structure 3DLQ containing the IL22-IL22Receptor complex was first glycosylated with large representative glycans (A4S4) using the glycoprotein builder in GLYCAM-Web (http://glycam.org), then visualized in VMD and represented as surface or CPK representations. The molecular model was imported into Cinema4D for lighting, texturing and rendering. Final rendered images were color-adjusted in Adobe Photoshop. The augmented reality visual in the supplemental was created using Adobe Aero.

### Data analysis
**Mass deconvolution.** Mass spectra from different DIA-PTCR scans were collected as a single raw file with individual scans corresponding to different DIA isolation windows. To deconvolve and assemble this data, the mass spectra from a raw file were loaded into UniDec with the UniChrom tool. Mass spectra were summed in 0.5 min non-overlapping windows using the time partitioning mode in UniChrom. For example, this approach assembled 100 summed mass spectra from the approximately 50 min acquisition.

Each spectrum was independently deconvolved by UniChrom using standard high-resolution native presets with slight modifications, including turning off normalization of data, limiting the mass range to 90–120 kDa, and setting the charge smooth width to −1, which uses a boxcar smoothing of charge states. We found that the −1 setting for deconvolution yielded fewer artifacts than the standard 1 setting, which uses a different smoothing function.

Following deconvolution, zero-charge mass spectra were summed into a composite spectrum. Peaks were detected from this composite spectrum. It is possible in UniDec to pick peaks on a per-spectrum basis using the Experimental > Get Scan Peaks function. However, we found the composite scans sufficient.

**Peak assignments.** Peaks were matched to potential glycoforms using the UniDec Python API functions and custom code. Example scripts are provided for each step at the UniDec GitHub page in the PublicScripts/DIA-PTCR folder. Initial attempts were made to match the peaks to protein complex mass plus individual combinations of glycans/PTMs identified from bottom-up proteomics. However, three challenges limited this approach. First, for several samples, like ovalbumin, the bottom-up proteomics data did not sufficiently cover the DIA-PTCR peaks. Many peaks did not have any potential combinations that matched, revealing proteoforms not detected in the bottom-up proteomics data. Second, for other samples, like IL22-Fc, the bottom-up proteomics data was too complex. For example, the IL22-Fc SA4 sample had 36, 33, and 13 potential glycans attached at three independent sites. Combining three sites per monomer into a dimer complex yields six sites with 238 million potential combinations. Although it is technically possible to simulate these potential combinations, inclusion of the fourth site with partial occupancy was computationally impractical. Finally, of these complex combinations, many were isomers with identical masses.

**Monosaccharide annotation.** To address these three challenges, we first developed a simpler search against combinations of monosaccharide building blocks rather than full glycans. We assumed the mass of the protein was fixed, and the number of core M3 glycans was also fixed. The numbers of fucoses, sialic acids, GlcNAcs, and hexoses were allowed to vary, and the limits were informed by both the bottom-up proteomics data and knowledge of glycan biology. For example, with IL22-Fc, the maximum number of fucose units was set to six because we expect a maximum of one fucose per glycan, and there are six primary sites. The numbers of sialic acids and GlcNAcs were set to 32, a max of 5 per glycan site. The hexoses were limited to below 48, a max of 8 per glycan. This simplified combination of glycans brought down the search space by three orders of magnitude to 373,527

possible combinations. These possible combinations were further filtered by the knowledge of glycan biology to enforce that the number of sialic acid units was less than or equal to the number of hexose units and less than or equal to the number of GlcNAc units. Using this glycan filtering limited the possible combinations in this example to 150,535. Combining this smaller search space with backend improvements to the UniDec code, this combinatorial set could be created and matched with the data in less than a second.

However, the improved matching approach did not solve the problem of multiple combinations with nearly the same masses. For example, four GlcNAc units ($4 \times 203.195 = 812.78$ Da) are very close to five hexoses ($5 \times 162.142 = 810.71$ Da). Similarly, two fucoses ($2 \times 146.143 = 292.29$ Da) are very close to one sialic acid (291.248 Da). In this relatively large search space, many possible combinations give very similar masses. In the IL22-Fc data, each peak had around 50–100 possible unique monosaccharide subunit combinations within a ± 5 Da (~50 ppm) tolerance. Assuming an equal probability of every glycan that matched, the summed distributions for each monosaccharide were very broad. Narrowing the tolerance to 0.5 Da (~5 ppm) reduced the number of potential assignments to 5–20, but it did not significantly change the distributions.

**Probabilistic monosaccharide annotation.** To help address the problem of hundreds of possible monosaccharide combinations for each peak, we integrated the DIA-PTCR data with glycoproteomics data. First, we decomposed the glycans identified by proteomics into the number of each monosaccharide subunit matched above, which the core M3 removed. For example, the glycan A2S1G1F was decomposed into 1 sialic acid, 2 GlcNAcs, 2 hexoses, and 1 fucose. Second, for each glycosite, the probability of finding any particular glycan was calculated by dividing the peak area measured by bottom-up proteomics over the total summed peak areas from all possible glycoforms at that site. Finally, we summed all possible combinations of each glycoform and combined their probabilities to find the probability of finding any given quartet of total monosaccharides ($n_{SA}$, $n_{GN}$, $n_{Hex}$, $n_{Fuc}$). Adding up the probabilities of finding each number of each monosaccharide yielded distributions of monosaccharide counts purely from the bottom-up proteomics data (Supplementary Fig. 12).

Informed by the bottom-up proteomics probabilities for each monosaccharide quartet, we then revisited the matching results. For each peak, all possible matches were found within a window of 5 Da. The probability of each match in the overall protein was estimated by multiplying the probability of finding that unique monosaccharide quartet (described above) by the peak height. To get the overall distributions of each monosaccharide, the probabilities for all peaks and all possible matches with a particular count (such as 2 sialic acids) were summed and normalized. This yielded a distribution of total monosaccharide counts informed by both the DIA-PTCR and the bottom-up proteomics (Fig. 4d–i, Supplementary Fig. 12). Here, the intensities are fundamentally derived from the DIA-PTCR data, but the assignments are statistically guided by the most probable monosaccharide combinations, as determined from the bottom-up proteomics data. Additional results are shown where the monosaccharide counts were isolated for each peak and thus show the distribution of potential monosaccharide counts as a function of mass of the intact protein (Fig. 4d–i).

Interestingly, statistical monosaccharide assignments for IL22-Fc yielded very similar results to the monosaccharide distributions from purely statistical combinations of the glycoproteomics data, revealing good agreement between the two data sets (Supplementary Figs. 12, 14). In other samples, such as MHCII, the bottom-up proteomics data initially only covered a small fraction of the peaks observed by DIA-PTCR. Addition of the truncated species was required to match all the peaks in the spectrum.

**Probabilistic glycan annotation.** After examining the monosaccharide distributions, we then revisited the glycan assignments using a probabilistic approach. For our discussion, we will use IL22-Fc SA4 as an example to illustrate the results. Based on the bottom-up proteomics data, there were eight sites on the dimer, but the last two sites are largely free. If we consider only 6 possible sites, there are a total of 238 million potential glycoforms. We then brute force calculated the probabilities of all these possible combinations and matched them with measured peaks, which took about 2 h on a standard computer.

These findings expose the staggering complexity and heterogeneity of glycoproteins. The most abundant peak in the IL22-Fc SA4 data has a mass of 98,224 Da. Within a window of ±5 Da, there are 614,270 potential matches to this single peak. This is significantly higher than the number of potential monosaccharide annotations because many combinations have the same total monosaccharide composition. If we consider the probability distribution for these potential assignments, the most likely assignment only accounted for 0.045% of the peak intensity. Interestingly, the two most likely forms (isomers with 2 × A4G4F, 2 × A4G0F, A3G0, A3S1G2F) were about 9 orders of magnitude more likely than the least probable assignment (A1G1F, A1G0F, A4S2G2F, A4S2G2F, A2S1G1, A2S1G1).

The 4th IL22-Fc glycosylation site is about 50% occupied, based on proteomics data, with a smaller set of potential glycans. Including the 4th site brings the potential search space to over 15 billion combinations, which is impossible to search on a standard computer due to memory limitations. However, inclusion of the 4th site is possible if we limit the search to only glycans with a relative site occupancy greater than 3% for each of the four monomer sites (67 million combinations). Here, similar results are observed, with 158,000 potential glycan combinations. The most likely annotations along with their probabilities are shown for the 98,224 Da peak in Supplementary Fig. 15.

To generate a "glycan barcode", the probabilities for each potential glycan on each potential site were summed across all peaks, with the relative probability for each potential peak assignment multiplied by the relative peak height from the DIA-PTCR data. Each of these was created from the 8-site glycoproteomics data, with a 3% threshold used for the SA4 and SA8 data and a 1% threshold used on the SA15 data. Smaller thresholds were not possible due to memory limitations. Similar barcodes were made for the DIA-PTCR data alone, which assumed an equal probability for all possible peak assignments that matched within ± 5 Da, and from the glycoproteomics data alone, without including peak heights from the DIA-PTCR data.

### Reporting summary
Further information on research design is available in the Nature Portfolio Reporting Summary linked to this article.

### Data availability
The DIA-PTCR data generated in this study have been deposited in the MassIVE database under accession code MSV000092002 Source data are provided as excel tables in Supplementary information Source data are provided with this paper.

### Code availability
Code for the integration of glycopeptide data, glycomics data and DIA-PTCR data is available at: https://github.com/michaelmarty/UniDec/tree/master/PublicScripts/DIA-PTCR.

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

## Acknowledgements

We would like to acknowledge Ania Hupalowska for valuable input regarding figures and visuals. We are also thankful to the Microchemistry, Proteomics and Lipidomics department at Genentech and Thermo Fisher Scientific for valuable insights and technological support provided during the development of the project. We would like to acknowledge funding from National Science Foundation grant CHE-1845230.

## Author contributions

C.M., J.D.H., R.D.M., J.E.P.S and R.H. aided in modifying the instrumentation, experimental design, and data acquisition, L.F.S. and W.S. prepared the paper, L.F.S., W.P., M.I.B., T.B., P.D., C.T., S.S., T.B., D.D., N.A., M.S., J.G., A.E., M.T.M. and W.S. performed described experiments and data analysis.

## Competing interests

L.S., W.P., M.I.B., T.B., P.D., C.T., S.S., T.B., D.D., J.G., A.E., N.A., M.M. and W.S. are employed by Genentech, Inc., a for-profit company that produces and markets therapeutics. C.M., J.H., J.E.P.S., R.M. and R.H. are employed by Thermo Fisher Scientific, a for-profit company that manufactures and sells mass spectrometry equipment. The remaining authors declare no competing interests.
