## [Peer Review File · Nature Communications]

Reviewers' Comments:

Reviewer #1:

Remarks to the Author:

The study presented by Luis et. al outlines the utility of native mass spectrometry in characterizing proteins with complex glycosylation. The technology of proton transfer charge reduction (PTCR), combined with quadrupole isolation and data deconvolution, provides a promising platform to delineate the macro-heterogeneity of protein glycosylation. Overall, the authors demonstrated a very feasible analytical approach, from the new DIA-PTCR data acquisition workflow with quadrupole isolation enabled by the Orbitrap Ascend to the use of UniDec for data analysis. It described how the mass contributed by complex glycosylation can be deconvoluted from the PTCR MS2 data to allow probabilistic glycan annotation, guided by the most probable monosaccharide combinations determined from bottom-up (glyco)proteomics data. The results demonstrated a significant improvement of mass spectrometry analysis in dissecting proteins that bear multiple glycosylation sites and glycoforms, especially under non-denaturing conditions while preserving the protein assembly.

1. It should be noted that the ideas of using PTCR, the extra functionalities afforded by the Orbitrap Ascend platform, and the publicly available UniDec software, are not exactly new, nor introduced for the first time. With the help of both hardware and software developers (included in the authorship), the authors have nonetheless nicely integrated these various technical aspects into a concerted workflow applied to a few multimeric glycoproteins of interest. In optimizing the data acquisition parameters, the authors experimented with PTCR reaction time and isolation using ion trap vs quadrupole, which is only available on the latest Orbitrap instrument. The advantage is obvious but the authors did not say much about the limitation without these advanced features. Can one infer that without using the quadrupole for isolation and the extended 16000 m/z maximum mass range, the analysis of ligand hexamer, MHCII and IL22-Fc as presented in this manuscript would not be possible? What would one get instead? What exactly are the enabling new features introduced here?

2. Data acquisition aside, the main challenge is still in obtaining a reproducible and confident glycoform assignment, given the inability to achieve accurate mass measurement at sufficient resolution. Their approach to glycan annotation and the results are seemingly believable but remain to be validated. Reproducibility of the complex data is an issue that needs to be convincingly shown if this top-down approach is to be used as a primary tool to monitor glycoprotein-based biotherapeutics. In this report, reproducibility and accuracy in mass measurement and hence its ensuing glycan annotation were only benchmarked using ovalbumin, and not reported for the "real samples".

3. Suppl Fig 4g: The Figure legend noted as "Number of monosaccharides per glycoforms plotted vs their probability of occurrence" but the axis in the Fig was labeled as "Number of glycan units per glycoforms vs Probability" - glycan units refer to monosaccharides? In all other Figures, this is noted as the "Number of monosaccharides per glycoforms plotted against their relative abundance". Is "relative abundance" similar to "probability"? This is a recurring issue throughout the manuscript.

4. For "Probabilistic Glycan Annotation" as described in the Methods section, it is unclear how the "probability distribution" for the potential assignments was calculated and how this value is related to "peak intensity" in the statement saying that "the most likely assignment only accounted for 0.045% of the peak intensity". One may speculate how the authors arrived at this conclusion, but it should be more transparently explained. Moreover, when referring to Suppl Fig 12, the relative abundance of the highest abundance glycoform composition described is 0.034%. How is this correlated with the 0.045% above? Are they referring to the same or different things? Altogether, although the big picture and main conclusion are straightforward, the description of the

probabilistic calculation and relative abundance is confusing and difficult to follow. The Supplemental Spreadsheets are not properly referenced and named, and are completely without explanatory note, which largely deter readers from navigating the data meaningfully.

5. For generating "glycan barcode", again, it is not easy to follow the statement "the probabilities for each potential glycan on each potential site were summed across all peaks, with the relative probability for each potential peak assignment multiplied by the relative peak height from the DIA-PTCR data" although one can more or less guess how this was done. What does it mean by the 3% or 1% threshold used for the glycoproteomics data?

6. Other few minor points:

- The glycan annotations from Fig. 1i, Fig. 3a, and Fig. 4b are too small and should be enlarged for better visibility.
- The results from UniDec deconvolution are very busy in all figures, and it is difficult to visualize the mass heterogeneity and its corresponding glycoforms. More graphic illustrations or summary tables could help the readers obtain more information from the results.
- According to the description on page 9, MHCII might have O-linked glycosylation. Did the authors observe this from their MS characterization and annotations?
- According to the glycan barcodes shown in Fig. 3, it appears that most of site 4 is unoccupied by glycan. Is this known from previous studies?
- In Fig. 4a, why is the scale of relative abundance not 1?

7. Three biotherapeutics were examined and showed different degrees of mass heterogeneity. It seems not to fully correlate with the number of glycosylation sites, as the TNF-Fc-VHH chimeric protein, with 6 N-glycosites, presents the most homogeneous size distribution compared to other biotherapeutics. Have the authors considered what the determining factors for the heterogeneity of glycosylation would be? Having a comprehensive dataset from glycomics and glycoproteomics analysis, it might be worth discussing this to provide readers with a better understanding of the complexity of protein glycosylation.

Reviewer #2:

Remarks to the Author:

The manuscript by Schachner et al describes a new approach to facilitate analysis of the highly heterogeneous glycosylation of biotherapeutics. While the current state-of-the-art is a laborious analysis of released glycans or proteolytic glycopeptides of a biotherapeutic by mass spectrometry (MS), this new approach claims to rather rapidly analyze the glycosylation of intact glycoproteins, without any denaturation, digestion or separation. It relies on glycoform fingerprinting using proton-transfer charge-reduction (PTCR) with gas-phase fractionation in a form of data-independent (DIA) tandem MS, followed by extensive bioinformatics and correlation with glycoproteomics and glycomics data. The power of the method is demonstrated on several different glycoproteins, each with different intrinsic complexity and heterogeneity. Initially, DIA-PTCR was used to analyze the glycosylation of a bispecific TNFL-hexamer-Fc with VHH domain at C' of one of the two Fc chains, that had 6 N-glycans (one on each TNFL unit). This approach could easily demonstrate the heterogeneity of glycosylation, as well as subunit mis-assembly (i.e. partial assembly or construct without VHH), something that could not have been achieved using the traditional MS techniques of fragmented samples. Furthermore, reproducibility is further demonstrated using ovalbumin. Next, DIA-PTCR was used to analyze peptide-bound MHCII containing 4 N-linked glycosylation sites at different parts of the molecule. As with the TNFL-Fc-VHH construct, the method could differentiate intact protein versus its truncated form, and broad heterogeneity of glycosylation. The data generated 'monosaccharide fingerprint' that was integrated with further glycoproteomic data to enable generation of 'glycan barcodes' that are site-resolved glycan composition and their predicted relative abundances. Finally, DIA-PTCR was used to analyze IL22-Fc that contain 8 N-linked glycosylation sites (4 on each of the IL-22 domains), and seemed to be highly heterogeneous compared to the other analyzed glycoproteins, thus required further method optimization yielding the glycan barcode with a glycosylation site resolved map, that also fit the glycomics data. The analysis revealed different sialylation variants, that were then functionally analyzed through binding to the IL22 receptor by ELISA, including relevant

mutants in different glycosylation sites. This DIA-PTCR analysis with the follow up biological investigation revealed that high sialylation, particularly at Asn21 site, abrogated binding to the IL22 receptor, and its reciprocal mutation N21Q led to ~4-fold increase in binding. Models of the glycosylated IL22 and the receptor-bound IL22 are shown to further support the conclusions. Overall, the manuscript is well-written and referenced, and the method is sound and convincingly seems to provide important innovation to this very complex task to rapidly analyze glycosylation of biotherapeutics. In particular, the ability to analyze intact glycoproteins adds a new level of information to this type of analysis that could not have been achieved with current techniques, and required indirect methodologies to infer such information. This approach is of potential great significance to the whole field of biotherapeutic design and quality assurance, and could potentially open up new ways to investigate the role of glycosylation in the fine tuning of biotherapeutic efficacy in vitro and in vivo.

Major comments:

1. The method is claimed to detect mis-assembled constructs in the TNF-ligand hexamer and the truncated alpha chain in peptide-bound MHCII, it would be useful to validate these claims also by other techniques
2. It was not clear which glycomics and glycoproteomics data were integrated into the full final analysis of the DIA-PTCR. If the data had been generated particularly for these proteins, then further details need to be provided for methodology and data.
3. Although macro-heterogeneity was mentioned (when not all site are fully occupied/glycosylated), it seems important to explain how this is addressed in methodology.

Minor comments:

1. List all abbreviations
2. In Supplemental Figure 2b define the nomenclature of glycosylation for clarity (e.g. A2G0F etc) or refer to a relevant supplementary file.
3. Supplemental Figure 3 correct numbering of the listed Figure 1j. In addition, it is not clear whether the table refers to the listed peaks in the inset, and why the units are different (units in Table are in Da but in Figure 1i it is in kDa). It would be useful to clarify how the mass shifts relative to the central peak allow the Glycan annotations (show/describe an example of one of the peaks)
4. Supplemental Figure 4 e and f – unify X-axis kDa numbering format
5. Supplemental Figure 5 a – correct units
6. Figure 3 a: IL-22-Fc is stated to contain 8 glycosylation sites in main text, however in the figure only 7 sites are shown.
7. In page 15: “4, 8 and 15 mol/mol sialic acid molar content”, and in Figure 3 d and Supplemental Figure 8 - define abbreviation sialic acids (SA).
8. The term potency is misleading with regard to IL22, perhaps replace with “binding to its receptor” or “in vitro potency of binding to its receptor”
9. In methods: Define the sequences of the P2A self-cleaving peptide sequence, flexible glycine/serine linker, signal sequence for secretion, Fos /Jun zipper
10. In methods/Figure 4d-e: IL22-Fc Potency by Binding Assay – clarify how many replicates were tested and how many times the exp was done

Reviewer #3:

Remarks to the Author:

REVIEWER COMMENTS

Reviewer #1 (Remarks to the Author):

The study presented by Luis et. al outlines the utility of native mass spectrometry in characterizing proteins with complex glycosylation. The technology of proton transfer charge reduction (PTCR), combined with quadrupole isolation and data deconvolution, provides a promising platform to delineate the macro-heterogeneity of protein glycosylation. Overall, the authors demonstrated a very feasible analytical approach, from the new DIA-PTCR data acquisition workflow with quadrupole isolation enabled by the Orbitrap Ascend to the use of UniDec for data analysis. It described how the mass contributed by complex glycosylation can be deconvoluted from the PTCR MS2 data to allow probabilistic glycan annotation, guided by the most probable monosaccharide combinations determined from bottom-up (glyco)proteomics data. The results demonstrated a significant improvement of mass spectrometry analysis in dissecting proteins that bear multiple glycosylation sites and glycoforms, especially under non-denaturing conditions while preserving the protein assembly.

1. It should be noted that the ideas of using PTCR, the extra functionalities afforded by the Orbitrap Ascend platform, and the publicly available UniDec software, are not exactly new, nor introduced for the first time. With the help of both hardware and software developers (included in the authorship), the authors have nonetheless nicely integrated these various technical aspects into a concerted workflow applied to a few multimeric glycoproteins of interest. In optimizing the data acquisition parameters, the authors experimented with PTCR reaction time and isolation using ion trap vs quadrupole, which is only available on the latest Orbitrap instrument. The advantage is obvious but the authors did not say much about the limitation without these advanced features. Can one infer that without using the quadrupole for isolation and the extended 16000 m/z maximum mass range, the analysis of ligand hexamer, MHCII and IL22-Fc as presented in this manuscript would not be possible? What would one get instead? What exactly are the enabling new features introduced here?

We thank the reviewer for their thoughtful comments, and true, these platforms independently have existed. Due to advancements in instrumentation via the commercially available Ascend platform we are now able to integrate in-line charge reduction, high m/z ion isolation, and stepped acquisition, which is absolutely the main innovation in this manuscript.

Regarding advantages of high m/z quadrupole isolation, we have amended the paragraph on pages 10-11 to provide further details. The new paragraph reads: "For research purposes, we investigated the analytical advantages of extending the quadrupole m/z filter precursor selection range to 8000 m/z (by lowering the drive frequency), and compare the performance in this extended mass range mode to precursor isolation in the RF linear ion trap analyzer (**Supp. Fig. 7**). Employing the high m/z quadrupole precursor selection proved advantageous in decreasing the number of proteoforms present in the individual stepped m/z selection windows relative to ion trap m/z isolation and provided

more interpretable MS2 spectra with higher product ion signal to noise ratio and less deconvolution ambiguity. In the analysis of other heterogeneous biotherapeutics over the past year we assert that inclusion of the higher resolution quadrupole precursor m/z selection into the workflow is absolutely necessary for mass interpretation of spectrally dense data.”

Ovalbumin (Supp Fig 4) and Hexamer (Fig 2) DIA-PTCR data acquisitions were all performed using ion trap precursor isolation, available on standard Eclipse and Ascend instruments. For biotherapeutics of moderate complexity (such as the molecules listed above) the ion trap isolation is sufficient; for extremely complex molecules, such as IL22Fc, a smaller isolation window provided by the quadrupole was necessary for putative annotations but not so necessary for basic fingerprinting. Supp Fig 7 shows the ion trap vs quad comparison: a single m/z isolation window for IL22Fc illustrates that while both isolation methods provide proteoform data, the quadrupole is far superior in the ability to minimize ion congestion due to the much narrower m/z window.

2. Data acquisition aside, the main challenge is still in obtaining a reproducible and confident glycoform assignment, given the inability to achieve accurate mass measurement at sufficient resolution. Their approach to glycan annotation and the results are seemingly believable but remain to be validated. Reproducibility of the complex data is an issue that needs to be convincingly shown if this top-down approach is to be used as a primary tool to monitor glycoprotein-based biotherapeutics. In this report, reproducibility and accuracy in mass measurement and hence its ensuing glycan annotation were only benchmarked using ovalbumin, and not reported for the "real samples".

Reproducibility and ‘believability’ of the complex deconvolved spectra is indeed a critical consideration for this method to be widely adopted. For low heterogeneity such as in the case of ovalbumin, we performed quadruplicate measurements with high reproducibility, shown in Supp Fig 4. To assess the ability of the workflow to identify differences between replicates, such as in a batch-to-batch comparison, and without the benefit of multiple batches of IL22Fc we conducted a further experiment for the purposes of addressing this question. We compared a sialidase treated sample in which the outermost sialic acid sugars were removed to our original sample. The treated sample resulted in a shift of major peaks with a net loss of 6000 Da, indicating near complete desialylation and also lending confidence to the correctness of resultant deconvolved spectra between samples. Results of this follow up experiment are described in the main text on page 11 and also included as new supp figs 8 and 9: “The IL22-Fc molecule was compared before and after sialidase treatment resulting in removal the outermost sialic acid moieties that contribute to the proteoform mass diversity. An overall mass reduction of 6000 Da was observed, indicating both near complete removal of the twenty-two sialic acids as well as highlighting the ability of the DIA-PTCR workflow to detect changes in proteoform heterogeneity from one sample to the next (**Supp. Fig. 8**).

As an additional validation of the reproducibility of the DIA-PTCR workflow from data acquisition to mass deconvolution of complex biotherapeutics, we analyzed the sialidase-treated IL22-Fc in three technical replicates. This experiment produced three equivalent distributions of molecular weights with Pearson correlation coefficients of 0.95-0.98 (Suppl. Fig. 9).”

3. **Suppl Fig 4g:** The Figure legend noted as "Number of monosaccharides per glycoforms plotted vs their probability of occurrence" but the axis in the Fig was labeled as "Number of glycan units per glycoforms vs Probability" - glycan units refer to monosaccharides? In all other Figures, this is noted as the "Number of monosaccharides per glycoforms plotted against their relative abundance". Is "relative abundance" similar to "probability"? This is a recurring issue throughout the manuscript.

We have corrected the axes in this figure and all others. They now consistently refer to number of monosaccharides per glycoform vs Relative Abundance.

4. For "Probabilistic Glycan Annotation" as described in the Methods section, it is unclear how the "probability distribution" for the potential assignments was calculated and how this value is related to "peak intensity" in the statement saying that "the most likely assignment only accounted for 0.045% of the peak intensity". One may speculate how the authors arrived at this conclusion, but it should be more transparently explained. Moreover, when referring to Suppl Fig 12, the relative abundance of the highest abundance glycoform composition described is 0.034%. How is this correlated with the 0.045% above? Are they referring to the same or different things? Altogether, although the big picture and main conclusion are straightforward, the description of the probabilistic calculation and relative abundance is confusing and difficult to follow. The Supplemental Spreadsheets are not properly referenced and named, and are completely without explanatory note, which largely deter readers from navigating the data meaningfully.

Agreed that the description of probabilistic calculations is quite confusing. We have attempted to remedy this confusion and also emphasize the ability of the method to render significant amounts of proteoform information, albeit non-complete in the case of IL22Fc, rather than focusing on strategies for assignments of peaks to glycoforms. In the text we have separated the description and demonstration of the DIA-PTCR method, as applied to MHCII (Fig 2) and IL22Fc (Fig 3), from the optional bioinformatic analysis of the datatypes.

Our main goal is to explain the universality of the method and we agree the annotation is obscuring that basic point. In addition to further explaining what can be understood from the stitched, deconvolved spectra we have made notes throughout the text to discuss the potential of the method on other sources of heterogeneity besides glycosylation.

On page 4 we have added the following statement: "DIA-PTCR resultant spectra may be interrogated as single scans to qualitatively evaluate the presence or absence of

individual proteoforms in the complex mixture or may be stitched together and deconvolved to demonstrate the entire proteoform landscape. To more quantitatively assign proteoforms such as glycan composition, one bioinformatic strategy we developed is to analyze the DIA-PTCR datatype first after stitching together spectra within the UniDec³³ software package to calculate the glycoform molecular weight distribution of the biotherapeutic. Next, the detected intact molecular weights are correlated with information from glycoproteomics and glycomics analyses, if available. While the distribution of molecular weights determined from the DIA-PTCR data defines the degree of heterogeneity of the glycoprotein and the stoichiometry of a non-covalent protein complex, insight into the specific post-translational modifications of the proteoforms may be limited.” On page 11 we added “Finally, we were interested in testing the ability of the DIA-PTCR method to assess batch-to-batch reproducibility. We thus fractionated and enriched IL22-Fc samples for 4, 8 and 15 mol/mol sialic acid molar content (**Figure 3d, Supp. Fig. 10**) as a surrogate to multiple product batches since only a single large-scale purification was available. We detected extensive differences in the molecular weight distributions of the three separate samples that were consistent with increasing amounts of sialylation.”

Finally we reorganized the glycoform assignment strategy in its own subsection with a separate figure in order to clarify for the reader that this is an optional part of the workflow and not necessary to obtain useful information about the composition of their sample. (page 12 and figure 4)

5. For generating "glycan barcode", again, it is not easy to follow the statement "the probabilities for each potential glycan on each potential site were summed across all peaks, with the relative probability for each potential peak assignment multiplied by the relative peak height from the DIA-PTCR data" although one can more or less guess how this was done. What does it mean by the 3% or 1% threshold used for the glycoproteomics data?

Anything below 3% or 1% relative site occupancy was not included as potential glycan for search simplicity, we have included the following clarification in the methods section: “To generate a “glycan barcode”, the probabilities for each potential glycan on each potential site were summed across all peaks, with the relative probability for each potential peak assignment multiplied by the relative peak height from the DIA-PTCR data. Each of these was created from the 8-site glycoproteomics data, with a 3% threshold used for the SA4 and SA8 data and a 1% threshold used on the SA15 data. Smaller thresholds were not possible due to memory limitations. Similar barcodes were made for the DIA-PTCR data alone, which assumed an equal

probability for all possible peak assignments that matched within ± 5 Da, and from the glycoproteomics data alone, without including peak heights from the DIA-PTCR data.”

6. Other few minor points:

- The glycan annotations from **Fig. 1i, Fig. 3a, and Fig. 4b** are too small and should be enlarged for better visibility.

Corrected, thank you!

- The results from UniDec deconvolution are very busy in all figures, and it is difficult to visualize the mass heterogeneity and its corresponding glycoforms. More graphic illustrations or summary tables could help the readers obtain more information from the results.

Agreed, we have revised all figures for simplicity and included supplemental spreadsheets detailing UniDec results should the reader wish to dig in.

- According to the description on page 9, MHCII might have O-linked glycosylation. Did the authors observe this from their MS characterization and annotations?

Thank you, we have added mention of O-linked glycosylation to the MHCII text and also to the glycan barcode description in Fig 4 legend

- According to the glycan barcodes shown in Fig. 3, it appears that most of site 4 is unoccupied by glycan. Is this known from previous studies?

Yes, this was also seen in the glycopeptide data (see supplemental spreadsheets containing glycoproteomic data). Bottom up site occupancy data shows <60% unoccupied for non-consensus glycan site (NAC). Please note that glycan barcodes are now displayed in the new main text Fig 4

- In Fig. **4a**, why is the scale of relative abundance not 1?

For legibility purposes we reported these data as fractional abundance. Together, all values for each dataset (SA4, 8 or 15) will add up to 1.

7. Three biotherapeutics were examined and showed different degrees of mass heterogeneity. It seems not to fully correlate with the number of glycosylation sites, as the TNF-Fc-VHH chimeric protein, with 6 N-glycosites, presents the most homogeneous size distribution compared to other biotherapeutics. Have the authors considered what the determining factors for the heterogeneity of glycosylation would be? Having a comprehensive dataset from glycomics and glycoproteomics analysis, it might be **worth discussing** this to provide readers with a better understanding of the complexity of protein glycosylation.

The reviewer is correct in that even though the number of occupied glycosylation sites is similar for TNF-Fc_VHH and IL22Fc, the glycoform heterogeneity of the samples are quite different. For example, TNF-Fc-VHH has three (identical) repeat TNF constructs attached to each of two Fc chains. As each of the glycosylation sites for this hexameric moiety is the same we predict the microheterogeneity to be minimal. In contrast, for IL22Fc there are eight potential glycosylation sites, four on each IL22 molecule. Of these four, three have been shown to be nearly fully glycosylated and each site contains a vast amount of microheterogeneity which was also seen in the orthogonal methods described.

Reviewer #2 (Remarks to the Author):

The manuscript by Schachner et al describes a new approach to facilitate analysis of the highly heterogenous glycosylation of biotherapeutics. While the current state-of-the-art is a laborious analysis of released glycans or proteolytic glycopeptides of a biotherapeutic by mass spectrometry (MS), this new approach claims to rather rapidly analyze the glycosylation of intact glycoproteins, without any denaturation, digestion or separation. It relies on glycoform fingerprinting using proton-transfer charge-reduction (PTCR) with gas-phase fractionation in a form of data-independent (DIA) tandem MS, followed by extensive bioinformatics and correlation with glycoproteomics and glycomics data. The power of the method is demonstrated on several different glycoproteins, each with different intrinsic complexity and heterogeneity. Initially, DIA-PTCR was used to analyze the glycosylation of a bispecific TNFL-hexamer-Fc with VHH domain at C' of one of the two Fc chains, that had 6 N-glycans (one on each TNFL unit). This approach could easily demonstrate the heterogeneity of glycosylation, as well as subunit mis-assembly (i.e. partial assembly or construct without VHH), something that could not have been achieved using the traditional MS techniques of fragmented samples. Furthermore, reproducibility is further demonstrated using ovalbumin. Next, DIA-PTCR was used to analyze peptide-bound MHCII containing 4 N-linked glycosylation sites at different parts of the molecule. As with the TNFL-Fc-VHH construct, the method could differentiate intact protein versus its truncated form, and broad heterogeneity of glycosylation. The data generated 'monosaccharide fingerprint' that was integrated with further glycoproteomic data to enable generation of 'glycan barcodes' that are site-resolved glycan composition and their predicted relative abundances. Finally, DIA-PTCR was used to analyze IL22-Fc that contain 8 N-linked glycosylation sites (4 on each of the IL-22 domains), and seemed to be highly heterogenous compared to the other analyzed glycoproteins, thus required further method optimization yielding the glycan barcode with a glycosylation site resolved map, that also fit the glycomics data. The analysis revealed different sialylation variants, that were then functionally analyzed through binding to the IL22 receptor by ELISA, including relevant mutants in different glycosylation sites. This DIA-PTCR analysis with the follow up biological investigation revealed that high sialylation, particularly at Asn21 site, abrogated binding to the IL22 receptor, and its reciprocal mutation N21Q led to ~4-fold increase in binding. Models of the glycosylated IL22 and the receptor-bound IL22 are shown to further support the conclusions.

Overall, the manuscript is well-written and referenced, and the method is sound and convincingly seems to provide important innovation to this very complex task to rapidly analyze glycosylation of biotherapeutics. In particular, the ability to analyze intact glycoproteins adds a new level of information to this type of analysis that could not have been achieved with current techniques, and required indirect methodologies to infer such information. This approach is of potential great significance to the whole field of biotherapeutic design and quality assurance, and could potentially open up new ways to investigate the role of glycosylation in the fine tuning of biotherapeutic efficacy in vitro and in vivo.

Major comments:

1. The method is claimed to detect mis-assembled constructs in the TNF-ligand hexamer and the truncated alpha chain in peptide-bound MHCII, it would be useful to validate these claims also by other techniques

Thank you for the suggestion! We have included the reduced and deglycosylated rpLC-MS analysis of the MHCII complex as Supp Fig 5 which confirms the truncation seen by DIA-PTCR.

2. It was not clear which glycomics and glycoproteomics data were integrated into the full final analysis of the DIA-PTCR. If the data had been generated particularly for these proteins, then further details need to be provided for methodology and data.

We have now included supplemental spreadsheets outlining the glycopeptide data we used to reduce the search space for the proteoform assignments, and also amended the methods section to include how these data were generated.

3. Although macro-heterogeneity was mentioned (when not all sites are fully occupied/glycosylated), it seems important to explain how this is addressed in methodology.

Thank you for the suggestion. Bottom up site occupancy data shows <60% unoccupied for the non-consensus glycan site (NAC) and is included in the supplemental glycopeptide spreadsheets.

Minor comments:

1. List all abbreviations

Abbreviation definitions now provided following the methods section.

2. In **Supplemental Figure 2b** define the nomenclature of glycosylation for clarity (e.g. A2G0F etc) or refer to a relevant supplementary file.

A supplemental spreadsheet describing glycan nomenclature is now provided

3. **Supplemental Figure 3** correct numbering of the listed Figure **1j**. In addition, it is not clear whether the table refers to the listed peaks in the inset, and why the units are different (units in Table are in Da but in Figure 1i it is in kDa). It would be useful to clarify how the mass shifts

relative to the central peak allow the Glycan annotations (show/describe an example of one of the peaks)

Corrected, thank you

4. **Supplemental Figure 4 e and f** – unify X-axis kDa numbering format

Corrected, thank you

5. **Supplemental Figure 5 a** – correct units

Corrected, thank you

6. **Figure 3 a:** IL-22-Fc is stated to contain 8 glycosylation sites in main text, however in the figure only 7 sites are shown.

The figure legend now clearly states that the 8th site is unoccupied - labeled with an empty box - for the glycoform represented in the illustration.

7. In page 15: “4, 8 and 15 mol/mol sialic acid molar content”, and in Figure 3 d and **Supplemental Figure 8** - define abbreviation sialic acids (SA).

Corrected, thank you, and SA is added to abbreviations list

8. The term potency is misleading with regard to IL22, perhaps replace with “binding to its receptor” or “in vitro potency of binding to its receptor”

Multiple changes on page 17 regarding the use of ‘potency’ were altered, thank you for the suggestion

9. In methods: Define the sequences of the P2A self-cleaving peptide sequence, flexible glycine/serine linker, signal sequence for secretion, Fos /Jun zipper

We have added the color-coded sequence below.

HHV1P-gD-SS_DPA1*02:02-linker-thrombin-Fos-linker-Avi-His6_P2A_HHV1P-gD-
SS-linker-CLIP-linker-TEV-linker-DPB1*05:01-linker-thrombin-Jun-linker-SpyTag003-linker-
His6

[viral signal seq]

GAIKADHVSTYAMFVQTHRPTGEFMFEFDEDEQFYVDLDKKETVWHLEEFGRAFSFEAQGGL
ANIAILNNLNTLIQRSNHTQAANDPPEVTVFVPELQGPNTLICHIDRFFPPVLNVTWLCNG
EPVTEGVAESLFLPRTDYSFHKFHYLTFVPSAEDVYDCRVEHWGLDQPLLKHWEAQEPIQMPE
TTESSADLVPRGSLTDTLQAETDQLEDEKSALQTEIANLLKEKEKLEFILAAAGSGSGSGLNDIFE

AQKIEWHEHHHHHGGGATNFLLKQAGDVEENPGP[viral
sigseq]GDSGT P VSKMRMATPLLMQA GGGGS ENLYFG GGGGSRATPENYLFQGRQECYAFN
GTQRFLERYIYNREELVRFSDVGEFRAVTELRPEAEYWNSQKDILEEKRAVPDRMCRHNYE
LDEAVTLQRRVQPKVNVSPSKKGPLQHNNLLVCHVTDFYPGSIQVRWFLNGQEETAGVVSTNL
IRNGDWFQILVMLEMTPQQGDVYICQVEHTSLDSPVTVEWKAQSDSARSKSSADLVPRGSR
ARLEEKVKTLKAQNSELASTANMLREQVAQLKQKVMNHGGSGGSRGVPHIVMVDAYKRYKG
GSHHHHHH

-
-
-

10. In methods/**Figure 4d-e**: IL22-Fc Potency by Binding Assay – clarify how many replicates were tested and how many times the exp was done

Methods and Figure 5d-e legend have been updated to reflect that the Binding assay was performed in duplicate.

Reviewer #3 (Remarks to the Author):

Reviewers' Comments:

Reviewer #1:

Remarks to the Author:

The authors have adequately addressed all issues raised. This is a much-improved manuscript.

Reviewer #2:

Remarks to the Author:

The authors properly addressed all concerns during the revision.

Reviewer #3:

Remarks to the Author:
